# Relationship between Participation in Daily Life Activities and Physical Activity in Stroke Survivors: A Protocol for a Systematic Review and Meta-Analysis

**DOI:** 10.3390/healthcare11152167

**Published:** 2023-07-30

**Authors:** Cristina de Diego-Alonso, Julia Blasco-Abadía, Almudena Buesa-Estéllez, Rafael Giner-Nicolás, María Pilar López-Royo, Patricia Roldán-Pérez, Víctor Doménech-García, Pablo Bellosta-López, Natalie Fini

**Affiliations:** 1Department of Physical Therapy, Faculty of Health Sciences, Universidad San Jorge, Campus Universitario, Autov. A23 km 299, Villanueva de Gállego, 50830 Zaragoza, Spain; cdediego@usj.es (C.d.D.-A.); jblasco@usj.es (J.B.-A.); abuesa@usj.es (A.B.-E.); rginer@usj.es (R.G.-N.); mplopez@usj.es (M.P.L.-R.); proldan@usj.es (P.R.-P.); vdomenech@usj.es (V.D.-G.); 2Department of Physiotherapy, Melbourne School of Health Sciences, University of Melbourne, Melbourne, VIC 3052, Australia; natalie.fini@unimelb.edu.au

**Keywords:** physical activity, participation, activities of daily living, stroke, relationship, protocol, systematic review, meta-analysis

## Abstract

Stroke survivors undertake low levels of physical activity and participation in daily life activities, but the correlation between these two domains still carries some degree of uncertainty. This systematic review and meta-analyses-based data synthesis will aim to describe and estimate the relationship between participation in daily life activities and physical activity in stroke survivors. Six databases (MEDLINE/PubMed, Web of Science, Scopus, PEDro, SPORTDiscus, and Rehabilitation & Sport Medicine Source) will be searched. Studies assessing participation alongside physical activity levels in adult stroke survivors in English or Spanish will be included. The study selection, assessment of the risk of bias, and data extraction will be conducted independently by two investigators. If available, correlation values between physical activity and participation outcomes will be extracted. The Hedges–Olkin method will be used for pooling correlation values between participation and physical activity measures. Subgroup analyses will be performed according to the time elapsed since the stroke (i.e., ≤6 months and >6 months). This will be the first systematic review with a meta-analysis to provide information on the relationship between physical activity and participation in stroke survivors. Findings are likely to inform the design of health prevention protocols and the development of healthy behavior change interventions.

## 1. Introduction

Stroke is the second leading cause of death in developed countries, and, according to World Health Organization (WHO) projections, one in four people will suffer a stroke throughout their lifetime [1]. Estimations indicate a significant rise in the number of stroke survivors worldwide by 2040–2050 [2,3,4,5,6,7,8]. Stroke is currently the third leading cause of disability worldwide [8] as it often leads to serious sequelae including sensory–motor, cognitive, and emotional impairments [9], which require neurorehabilitation [10]. For example, two-thirds show difficulty in walking independently, which is maintained beyond three months post-stroke [11]. These post-stroke sequelae generate limitations in activities and restrict participation in daily life activities such as social or leisure activities [12,13]. While immediate post-stroke sequelae might be inevitable, the subsequent sedentary lifestyle often observed in stroke survivors is believed to be modifiable [14]. Therefore, participation and physical activity (PA) are key targets for healthy lifestyles for stroke survivors.

Participation is defined by the International Classification of Functioning, Disability and Health (ICF) as a person’s involvement in life situations, which means an interaction between the individual’s health condition and contextual factors (i.e., environmental and personal factors) [15,16]. In addition, participation implies the performance of meaningful activities, habits, role performance, and social and community involvement [17]. Restoration of participation and functional levels is the main goal in occupational therapy [18]. However, the evidence regarding factors influencing the recovery of participation in daily life activities is limited. Commonly work, leisure, and domestic activities are the most affected after stroke [19,20].

PA is defined by the WHO as “any bodily movement produced by skeletal muscles that requires energy expenditure”; therefore, this includes all movement during leisure time, for transport to get to and from places, or as part of a person’s work and domestic duties [21]. PA can involve different intensities, such as light, moderate, and vigorous-intensity PA [21]. Following stroke, survivors are commonly inactive and have significantly increased sedentary lifestyles [22,23,24], spending long periods without activity, which may be related to the effects of the stroke [25]. In essence, post-stroke PA levels are lower than that recommended by the WHO for preventing diseases such as stroke [26].

It has been observed that a greater capacity for functional independence implies greater participation, with the ability to walk and drive, as well as having an extensive social network, being considered enablers [27]. However, it has been shown that despite increased physical functioning, most people continue to have participation restrictions in social and leisure [27], work [28], and housekeeping activities [29], which generates a high level of concern for stroke survivors [30].

A previous review recommended that studies must include several complementary assessment tools to globally record the level of participation and PA to establish a complete correlation between these variables [31,32]. To date, no systematic review has been conducted to synthesize the evidence on the relationship between participation and PA in stroke survivors. This review will be the first to provide key information to understanding the relationship between participation in daily activities and PA levels in the stroke survivor population.

A low level of PA is a primary risk factor for stroke recurrence, which, together with the level of participation in activities of daily living, is an indicator of the quality of life of stroke survivors. Additionally, a low level of PA is associated with a higher degree of dependency, a situation that impacts healthcare expenditure due to resource requirements [33]. Considering the WHO’s demand to seek solutions [34], it is imperative to update and synthesise the current information available and produce a meta-analysis on the relationship between PA levels and participation. To date, only a separate analysis of these data is available in the stroke population [12,35,36,37,38,39]. The results of this study will allow to understand whether people with higher levels of participation in activities of daily living are those who are less sedentary, or, on the contrary, if participating in activities of daily living does not correlate with reaching recommended levels of PA. Therefore, this systematic review with a meta-analysis may or may not demonstrate the need to target participation in activities of daily living, along with PA levels for secondary prevention in stroke survivors.

### Objectives

The general objective of this systematic review and meta-analysis is to provide a comprehensive understanding of whether there is a relationship between levels of PA and participation in individuals who have experienced a stroke. However, this review and meta-analysis is not intended to evaluate the effectiveness of interventions.

This systematic review also aims to comprehensively gather and synthesize all available scientific evidence reporting PA levels and participation in the population of stroke survivors. This will include a synthesis of the most used assessment tools and reported variables.

Additionally, the meta-analysis aims to quantitatively estimate potential correlations between participation in activities of daily living and PA among stroke survivors. The meta-analysis will pool data from different studies, providing a more comprehensive and representative overview of the research evidence. It will offer a confidence interval to determine the strength of the relationship between PA and participation, and it will help clarify any potential inconsistent or contradictory findings.

## 2. Materials and Methods

### 2.1. Study Design and Registration

This protocol has been reported following the Preferred Reporting Items for Systematic Review and Meta-Analysis Protocols (PRISMA-P) [40] and it was registered in the International Prospective Register of Systematic Reviews (PROSPERO) database (CRD42022360711).

Given the aim and observational nature of this systematic review and meta-analysis, the research question was formulated using the PEO framework (Population: stroke; Exposure: level of PA; Outcome: level of participation) to investigate the association between specific exposures and outcomes [41,42].

### 2.2. Eligibility Criteria

Table 1 presents a synthesis of the inclusion and exclusion criteria that will be employed during the study selection process.

#### 2.2.1. Study Types

Studies with a cross-sectional or prospective design evaluating participation alongside PA levels in stroke survivors will be included. Articles with full-text available and published in a peer-reviewed journal in English or Spanish languages will be included. Furthermore, if compatible results with the research question are identified in conference proceeding abstracts, the authors will be contacted to request the full data to ensure that relevant results related to the research question are not overlooked or excluded. The search for relevant studies will not have any restrictions based on the date of publication. However, to minimize the risk of bias resulting from small sample sizes during the meta-analysis, only studies with a sample size of more than 10 participants will be included [43].

#### 2.2.2. Participant Characteristics

Studies recruiting adult participants (≥18 years) with stroke diagnosis regardless of time since stroke or severity of stroke sequelae, the etiology, sex, or geographical location will be included. Studies with mixed populations with composite data will be excluded unless stroke data is able to be extracted separately. In addition, the population will be grouped according to time since stroke (less than or equal to 6 months and more than 6 months) for subsequent data synthesis and analysis.

#### 2.2.3. Characteristics of the Outcome Measures

Studies reporting PA data (e.g., minutes of PA at different intensities, caloric expenditure per PA, steps, sitting/standing time) measured by objective devices (e.g., pedometers, accelerometers), self-reported questionnaires or observational tools, and degree of participation in at least at least one area of daily living activity according to the occupational therapy framework: basic activities and instrumental activities of daily living, rest and sleep, education, work, play, leisure, or social participation [17] will be included (Table 2). Studies only including physical performance tests will be excluded, as these cannot be considered as direct indicators of PA.

### 2.3. Data Sources and Searches

A search will be carried out in six electronic databases (MEDLINE via PubMed; Web of Science and Scopus via Web of Science, managed by the Spanish Foundation for Science and Technology; SPORTDiscus and Rehabilitation & Sport Medicine Source via EBSCOhost; and PEDro) until April 2023. Search terms will be grouped into three categories: stroke-related words, PA terms, and participation in daily life activities terms. The different search strategies used in each database are presented in Table 3. Moreover, a thorough examination of the reference lists included in the full-text articles reviewed will be conducted.

To ensure a comprehensive and robust search strategy that encompasses all relevant aspects of the topic, synonyms of terms related to the three categories were included in the search strategy. Additionally, a separate analysis of search strategies employed in previous systematic reviews focusing on participation [12,44,45] and PA [14,22,46,47] was conducted, considering that this review will be the first to encompass both domains simultaneously. Furthermore, MeSH terms, concepts, and terms utilized in the Occupational Therapy framework [17] and the ICF [48] were incorporated.

### 2.4. Studies Selection

After removing duplicate records, the initial stage will involve screening studies based on their titles and abstracts, applying the predefined eligibility criteria. Subsequently, the full texts of the selected studies will be thoroughly reviewed and evaluated, again applying the same eligibility criteria. Studies will be selected at each step by 2 independent researchers (CDA and ABE), and, in case of disagreement, a consensus will be sought by a third researcher (PBL). The EndNote software will be used for reference management [49].

### 2.5. Evaluation of the Risk of Bias

Two researchers (CDA and PRP) will independently examine the risk of bias of the studies using the Newcastle–Ottawa scale for cohorts or case–control studies [50] and its adaptation to cross-sectional studies [51], while the PEDro scale will be used for clinical trials [52]. A third researcher (PBL) will verify the assessments and resolve discrepancies if any arise.

The Newcastle–Ottawa scale evaluates seven to eight items categorized into three criteria (selection, comparability, and exposure or outcome) with a maximum score of 9 (10 in the case of cross-sectional studies). Articles scoring at least 7 will be classified as “high quality”, 4–6 as “fair quality”, and less than 4 as “poor quality” [50].

The PEDro scale evaluates eleven items assessing the internal validity and interpretability of the results, where the maximum score is 10 (i.e., one point per item), as the first item is not considered in the final punctuation. Articles scoring at least 6 were considered of “high quality”, while a score of 4–5 was considered of “fair quality”, and less than 4 points was considered of “poor quality” [52].

Furthermore, to assess reporting bias, an initial step will involve comparing the measures analyzed and presented in the study’s protocol, if available. Additionally, the consistency between the variables described in the method section and those reported in the results section of the selected articles will be checked [53].

Finally, a qualitative analysis of the quality of evidence and the strength of recommendations will be conducted using the GRADE approach, similar to previous systematic reviews and meta-analyses of associations [54].

### 2.6. Data Collection

Two independent researchers (CDA and RGN) will extract data from the selected studies using a standardized data extraction sheet (Appendix A), while a third and fourth researcher (PBL and JBA) will create independent databases, verify that the extracted data match, and resolve any discrepancies.

The following data will be collected: general study information (title, authors, year of publication); sample and subgroup characteristics (sample size, mean age and sex, time since stroke, stroke severity); study characteristics (research design and country); outcome measures (timing of measurement, PA objective device, PA self-reported assessments, participation assessments); intervention, if any; outcome data (PA and participation). In addition, main results, including correlation findings between participation and PA outcomes will be extracted when possible. The results will be grouped and analyzed in relation to the variable of time elapsed since the stroke in the sample. In this way, there will be 2 groups, a sample of participants who have had a stroke less than or equal to 6 months ago and those who have had a stroke more than 6 months ago.

As this review and meta-analysis does not aim to evaluate intervention effectiveness, it is important to clarify that in the case of randomized controlled trials, the treatment arms will be treated as independent cohorts [55]. Consequently, data for each group will be extracted separately.

### 2.7. Data Synthesis

The data synthesis will be categorized based on the time since stroke, distinguishing between participants who experienced a stroke within 6 months of the study and those who had a stroke more than 6 months ago. This grouping will allow for a comparative analysis of outcomes and potential differences between these two subgroups of stroke survivors.

When there are at least two studies that report a correlation between the same participation and PA outcomes, the weighted summary of correlation coefficients under the random effects model will be calculated using the Hedges–Olkin method, based on a Fisher Z transformation of the correlation coefficients [56]. Correlations will be considered as ‘strong’ (ρ ≥ 0.70), ‘moderate’ (0.40 > ρ < 0.69), ‘weak’ (0.10 > ρ < 0.39), or ‘negligible’ (ρ < 0.10) [57]. If sufficient studies are found, a subgroup analysis will be performed according to areas of participation and/or PA measurement method (i.e., self-reported or objective measures). Additionally, depending on the quality, quantity, and homogeneity of the data obtained in the review, subgroup analyses may be conducted according to variables such as PA intensity or areas of participation. Moreover, age, sex, and/or stroke severity will be included in the analysis as moderators. Heterogeneity between study results will be investigated using I^2^ statistics with values > 50% indicating substantial heterogeneity across studies [58]. Publication bias will be examined by using funnel plots and Egger’s tests [59]. All analyses will be conducted using STATA v.16.1 (StataCorp, College Station, TX, USA), and alpha will be set at *p* < 0.05.

In cases where the data are not reported directly in an article, up to three attempts will be made to contact the study authors via email to obtain the data. By reaching out to the authors directly, the risk of missing valuable data is minimized, allowing for a more comprehensive analysis. However, if data are finally unavailable, the article will be excluded from the meta-analysis.

### 2.8. Deviations from PROSPERO Registry

This protocol provides updated information to address the shortcomings identified in the original PROSPERO registry. The registry erroneously indicated that interventions would be evaluated and did not mention the conduct of a meta-analysis. Due to the characteristics of the registry, it is not possible to amend the content beyond updating the review’s status. Therefore, this protocol aims to rectify these issues and provide accurate and comprehensive information for the systematic review and meta-analysis that will be conducted.

## 3. Expected Results

This systematic review aims to address the knowledge gap regarding the relationship between PA and participation levels in people following stroke. The research team anticipate that there will be a positive correlation between PA and participation. We anticipate a higher likelihood of finding a positive correlation in those more than 6 months post-stroke, considering the prolonged recovery process associated with participation [60].

The results of this systematic review will offer a comprehensive synthesis of existing research on the level of PA and participation after a stroke, taking into account the duration since the stroke. This review will also provide potential suggestions for methodological enhancements in future studies that explore the relationship between participation and PA, based on the analysis of methodological quality across all publications to date.

Aligned with the WHO’s priority guidelines for health and wellbeing by 2030, secondary prevention focuses on reducing the risk of recurrent strokes [34]. Addressing sedentary lifestyle is crucial, as it is associated with an increased risk of stroke recurrence and impacts an individual’s ability to participate in daily activities. By emphasizing the importance of participation and PA, this systematic review with meta-analysis aims to provide a robust foundation for informing clinical practices in neurorehabilitation. By considering the results according to the duration post-stroke, rehabilitation therapies could be better tailored to optimize outcomes and reduce health costs associated with dependence.

## Figures and Tables

**Table 1 healthcare-11-02167-t001:** Summary of eligibility criteria.

PEO	Inclusion Criteria	Exclusion Criteria
Population	Stroke survivors: >18 years, ≤6 months and >6 months since stroke. Study sample ≥ 10.	Transient ischemic attack, composite sample, or coexistence of other neurological disorders.
Exposition/ Outcome 1	Physical activity measures: recorded by objective diapositives and self-reported questionnaires.	Physical performance tests (e.g., 6 min walking test, 10 m walk test, treadmill exercise test).
Exposition/ Outcome 2	Participation measures: recorded by direct observational techniques and self-reported questionnaires.	Domains not covered by the occupational therapy framework.

**Table 2 healthcare-11-02167-t002:** Examples of potential tools and variables related to physical activity and participation outcomes that are anticipated to be identified while conducting the review.

Exposure/ Outcome	Type of Assessment	Potential Tools and Methods	Potential Variables
Physical activity	Objective devices	Pedometers Accelerometers Fitness trackers Smartwatches Mobile applications	METs h/day Total PA (Min/day) Vigorous PA (Min/day) Moderate PA (Min/day) Walking duration (Min/day) Steps/Day Sedentary time (sitting or lying) (Min/day)
	Self-reported questionnaires	International Physical Activity Questionnaire Physical Activity Scale for the Elderly	
Participation	Direct observation	Behavioral mapping	Total Participation level (0/100) Level of Basic ADL (0/100) Level of Instrumental ADL (0/100) Level of Social participation (0/100) Level of Work Participation (0/100) Level of Leisure Participation (0/100)
	Self-reported questionnaires	Stroke Impact Scale Barthel Index Frenchay Activities Index Activity Card Sort	

Abbreviations: ADL = activities of daily living; METs = metabolic equivalents; PA = physical activity.

**Table 3 healthcare-11-02167-t003:** Search strategies according to each database.

Database	Medline (Through PubMed)
Procedure	Combining the search strategy with the Boolean Operator: AND.
*Stroke*	(“Stroke”[Mesh] OR stroke[tiab] OR “cerebrovascular accident”[tiab] OR “cerebral vascular accident”[tiab] OR “cerebral vasospasm”[tiab] OR “cerebral bleed”[tiab] OR “post stroke”[tiab] OR poststroke[tiab] OR “cerebral ischaemia”[tiab] OR “brain ischemia”[tiab] OR “brain infarct”[tiab] OR “cerebral infarct”[tiab] OR “brain haemorrhage”[tiab] OR “cerebral haemorrhage”[tiab] OR “intracerebral haemorrhage”[tiab] OR “subarachnoid haemorrhage”[tiab] OR “intracranial haemorrhage”[tiab] OR “brain hemorrhage”[tiab] OR “cerebral hemorrhage”[tiab] OR “intracerebral hemorrhage”[tiab] OR “subarachnoid hemorrhage”[tiab] OR “intracranial hemorrhage”[tiab])
*Physical Activity*	(“Sedentary Behavior”[Mesh] OR “sedentary time”[tiab] OR “physical inactivity”[tiab] OR “physical activity”[tiab] OR walk* OR steps OR MVPA OR MPA OR “moderate to vigorous physical” OR LPA OR LIPA OR “light intensity physical activity” OR “metabolic equivalent of task” OR “Actigraphy”[Mesh] OR “activity monitor” OR “activity tracker” OR acceleromet* OR “pedometer” OR sensewear OR sense-wear OR activPAL OR “PAL 2” OR PAL2 OR IDEEA OR actigraph* OR actiwatch* OR stepwatch OR “actical” OR “activ8” OR “apple watch” OR “googlefit” OR “movemonitor” OR “step activity monitor” OR “StepWatch” OR “axivity” OR “motion logger*” OR motionlogger* OR fitbit OR IPAQ OR “International Physical Activity Questionnaire” OR PASE OR “Physical Activity Scale for the Elderly” OR PACE OR “Physician-based Assessment AND Counselling for Exercise”)
*Participation*	(“occupational area*” OR “daily life activities”[tiab] OR “daily life activities”[tiab] OR “everyday occupations”[tiab] OR ADL[tiab] OR “activities of daily living”[tiab] OR BADL[tiab] OR “basic activities of daily living”[tiab] OR IADL[tiab] OR “instrumental activities of daily living”[tiab] OR “self-care”[tiab] OR “leisure activities”[tiab] OR “social activities”[tiab] OR “community activities”[tiab] OR “community-dwelling”[tiab] OR “activities at home”[tiab] OR “social life”[tiab] OR sleep*[tiab] OR rest[tiab] OR “Occupational habit*” OR autonomy OR independence OR “occupational balance” OR “occupational satisfaction” OR “role performance” OR “community participation” OR “social participation” OR “return to work” OR “recovery activities” OR “meaningful activities” OR “participation restriction” OR “participation satisfaction” OR “role performance” OR “community reintegration” OR “community mobility” OR “behavioral mapping” OR ACS OR “activity card sort” OR SIS OR “stroke impact scale” OR BI OR “barthel index” OR FAI OR “frenchay activities index” OR “assessment of life habits” OR LHS OR “london handicap scale” OR LIFE-H OR “assessment of life habits” OR RNLI OR “reintegration to normal living”)
Filters	Language: English OR Spanish Species: Humans Publication date: All years available
**Database**	**Web of Science (through Web of Science, managed by the Spanish Foundation for Science and Technology)**
**Procedure**	Advanced search in Web of Science—All databases. Combining the search strategy with the Boolean Operator: AND.
*Stroke*	AB = (stroke OR “cerebrovascular accident” OR “cerebral vascular accident” OR “cerebral vasospasm” OR “cerebral bleed” OR “post stroke” OR poststroke OR “cerebral ischaemia” OR “brain ischemia” OR “brain infarct” OR “cerebral infarct” OR “brain haemorrhage” OR “cerebral haemorrhage” OR “intracerebral haemorrhage” OR “subarachnoid haemorrhage” OR “intracranial haemorrhage” OR “brain hemorrhage” OR “cerebral hemorrhage” OR “intracerebral hemorrhage” OR “subarachnoid hemorrhage” OR “intracranial hemorrhage”)
*Physical Activity*	AB = (“Sedentary Behavior” OR “sedentary time” OR “physical inactivity” OR “physical activity” OR walk* OR steps OR MVPA OR MPA OR “moderate to vigorous physical” OR LPA OR LIPA OR “light intensity physical activity” OR “metabolic equivalent of task” OR “Actigraphy” OR “activity monitor” OR “activity tracker” OR acceleromet* OR “pedometer” OR sensewear OR sense-wear OR activPAL OR “PAL 2” OR PAL2 OR IDEEA OR actigraph* OR actiwatch* OR stepwatch OR “actical” OR “activ8” OR “apple watch” OR “googlefit” OR “movemonitor” OR “step activity monitor” OR “StepWatch” OR “axivity” OR “motion logger*” OR motionlogger* OR fitbit OR IPAQ OR “International Physical Activity Questionnaire” OR PASE OR “Physical Activity Scale for the Elderly” OR PACE OR “Physician-based Assessment and Counselling for Exercise”)
*Participation*	AB = (“occupational area*” OR “daily life activities” OR “daily life activities” OR “everyday occupations” OR ADL OR “activities of daily living” OR BADL OR “basic activities of daily living” OR IADL OR “instrumental activities of daily living” OR “self-care” OR “leisure activities” OR “social activities” OR “community activities” OR “community-dwelling” OR “activities at home” OR “social life” OR sleep* OR rest OR “Occupational habit*” OR autonomy OR independence OR “occupational balance” OR “occupational satisfaction” OR “role performance” OR “community participation” OR “social participation” OR “return to work” OR “recovery activities” OR “meaningful activities” OR “participation restriction” OR “participation satisfaction” OR “role performance” OR “community reintegration” OR “community mobility” OR “behavioral mapping” OR ACS OR “activity card sort” OR SIS OR “stroke impact scale” OR BI OR “barthel index” OR FAI OR “frenchay activities index” OR “assessment of life habits” OR LHS OR “london handicap scale OR LIFE-H OR “assessment of life habits” OR RNLI OR “reintegration to normal living”)
Filters	Language: English OR Spanish Publication Years: All years available MeSH Headings: Humans
**Database**	**SCOPUS (through Web of Science, managed by the Spanish Foundation for Science and Technology)**
**Procedure**	Advanced document search. Combining the search strategy with the Boolean Operator: AND.
*Stroke*	TITLE-ABS(“Stroke” OR stroke OR “cerebrovascular accident” OR “cerebral vascular accident” OR “cerebral vasospasm” OR “cerebral bleed” OR “post stroke” OR poststroke OR “cerebral ischaemia” OR “brain ischemia” OR “brain infarct” OR “cerebral infarct” OR “brain haemorrhage” OR “cerebral haemorrhage” OR “intracerebral haemorrhage” OR “subarachnoid haemorrhage” OR “intracranial haemorrhage” OR “brain hemorrhage” OR “cerebral hemorrhage” OR “intracerebral hemorrhage” OR “subarachnoid hemorrhage” OR “intracranial hemorrhage”)
*Physical Activity*	TITLE-ABS(“Sedentary Behavior” OR “sedentary time” OR “physical inactivity” OR “physical activity” OR walk* OR steps OR mvpa OR mpa OR “moderate to vigorous physical” OR lpa OR lipa OR “light intensity physical activity” OR “metabolic equivalent of task” OR “Actigraphy” OR “activity monitor” OR “activity tracker” OR acceleromet* OR “pedometer” OR sensewear OR sense-wear OR activpal OR “PAL 2” OR pal2 OR ideea OR actigraph* OR actiwatch* OR stepwatch OR “actical” OR “activ8” OR “apple watch” OR “googlefit” OR “movemonitor” OR “step activity monitor” OR “StepWatch” OR “axivity” OR “motion logger*” OR motionlogger* OR fitbit OR ipaq OR “International Physical Activity Questionnaire” OR pase OR “Physical Activity Scale for the Elderly” OR pace OR “Physician-based Assessment and Counselling for Exercise”)
*Participation*	TITLE-ABS(“occupational area*” OR “daily life activities” OR “daily life activities” OR “everyday occupations” OR adl OR “activities of daily living” OR badl OR “basic activities of daily living” OR iadl OR “instrumental activities of daily living” OR “self-care” OR “leisure activities” OR “social activities” OR “community activities” OR “community-dwelling” OR “activities at home” OR “social life” OR sleep* OR rest OR “Occupational habit*” OR autonomy OR independence OR “occupational balance” OR “occupational satisfaction” OR “role performance” OR “community participation” OR “social participation” OR “return to work” OR “recovery activities” OR “meaningful activities” OR “participation restriction” OR “participation satisfaction” OR “role performance” OR “community reintegration” OR “community mobility” OR “behavioral mapping” OR acs OR “activity card sort” OR sis OR “stroke impact scale” OR bi OR “barthel index” OR fai OR “frenchay activities index” OR “assessment of life habits” OR lhs OR “london handicap scale” OR life-h OR “assessment of life habits” OR rnli OR “reintegration to normal living”)
Filters	Language: English OR Spanish Publication Years: All years available
**Database**	**Rehabilitation & Sports Medicine Source, and SPORTDiscus with Full Text (through EBSCOhost)**
**Procedure**	Advanced Search. Combining the search strategy (abstract) with the Boolean Operator: AND.
*Stroke*	stroke OR “cerebrovascular accident” OR “cerebral vascular accident” OR “cerebral vasospasm” OR “cerebral bleed” OR “post stroke” OR poststroke OR “cerebral ischaemia” OR “brain ischemia” OR “brain infarct” OR “cerebral infarct” OR “brain haemorrhage” OR “cerebral haemorrhage” OR “intracerebral haemorrhage” OR “subarachnoid haemorrhage” OR “intracranial haemorrhage” OR “brain hemorrhage” OR “cerebral hemorrhage” OR “intracerebral hemorrhage” OR “subarachnoid hemorrhage” OR “intracranial hemorrhage”
*Physical Activity*	“sedentary Behavior” OR “sedentary time” OR “physical inactivity” OR “physical activity” OR walk* OR steps OR MVPA OR MPA OR “moderate to vigorous physical” OR LPA OR LIPA OR “light intensity physical activity” OR “metabolic equivalent of task” OR “Actigraphy” OR “activity monitor” OR “activity tracker” OR acceleromet* OR “pedometer” OR sensewear OR sense-wear OR activPAL OR “PAL 2” OR PAL2 OR IDEEA OR actigraph* OR actiwatch* OR stepwatch OR “actical” OR “activ8” OR “apple watch” OR “googlefit” OR “movemonitor” OR “step activity monitor” OR “StepWatch” OR “axivity” OR “motion logger*” OR motionlogger* OR fitbit OR IPAQ OR “International Physical Activity Questionnaire” OR PASE OR “Physical Activity Scale for the Elderly” OR PACE OR “Physician-based Assessment AND Counselling for Exercise”
*Participation*	“occupational area*” OR “daily life activities” OR “daily life activities” OR “everyday occupations” OR ADL OR “activities of daily living” OR BADL OR “basic activities of daily living” OR IADL OR “instrumental activities of daily living” OR “self-care” OR “leisure activities” OR “social activities” OR “community activities” OR “community-dwelling” OR “activities at home” OR “social life” OR sleep* OR rest OR “Occupational habit*” OR autonomy OR independence OR “occupational balance” OR “occupational satisfaction” OR “role performance” OR “community participation” OR “social participation” OR “return to work” OR “recovery activities” OR “meaningful activities” OR “participation restriction” OR “participation satisfaction” OR “role performance” OR “community reintegration” OR “community mobility” OR “behavioral mapping” OR ACS OR “activity card sort” OR SIS OR “stroke impact scale” OR BI OR “barthel index” OR FAI OR “frenchay activities index” OR “assessment of life habits” OR LHS OR “london handicap scale” OR LIFE-H OR “assessment of life habits” OR RNLI OR “reintegration to normal living”
Filters	Language: English OR Spanish Publication Years: All years available
**Database**	**PEDro**
**Procedure**	Advanced search.
*Search (1)*	Abstract & Title: Stroke “Physical Activity” Subdiscipline: Neurology
*Search (2)*	Abstract & Title: Stroke Participation Subdiscipline: Neurology
Filters	Not applicable

## Data Availability

Not applicable.

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
