# Peer review of "Relationship between Participation in Daily Life Activities and Physical Activity in Stroke Survivors: A Protocol for a Systematic Review and Meta-Analysis"

_healthcare, 2023, doi:10.3390/healthcare11152167_

Round 1
Reviewer 1 Report
Thank you for the opportunity to review this manuscript. Below you will find some suggestions for improving the reporting of your protocol and some questions raised by reading it.
I believe the topic is relevant and up-to-date. However, the reporting of methods raises several questions that should be clarified.
Introduction
Ln 71-72: why is this synthesis important? Why is it key to understand the relationship? Please detail the relevance of this review. What is the advantage in performing meta-analysis?
Methods
Ln. 79-80: PRISMA is not a method, it was developed for reporting purposes. Please present the PICO question and the framework used to develop this protocol.
Ln. 81: the protocol registered in PROSPERO is not the same presented now.
Ln. 85: peer reviewed abstracts from conference proceedings will not be included? Authors should request full data from abstracts so as not to risk missing results relevant to the research question. Selection bias should be prevented as extensively as possible.
Ln. 88: please provide reason to exclude studies with sample sizes under 10 participants.
Distinguish primary outcomes measures and secondary outcome measures. Will there be any subgroup analysis according to PA intensity? Or other measure? Outcomes measures are very wide. How do authors plan to organize data?
Ln. 104-105: search in other sources must be included (such as search in reference list).
Ln. 108-109: which was the strategy to identify the search terms?
Ln. 114: Will reference software manager be use?
Ln. 114: eligibility criteria will not be applied during screening?
Ln. 119: How will risk of bias be assessed? How will reporting bias be assessed?
Ln. 127: how will authors deal with missing data? Will there be any subgroup analysis?
Ln. 146: Will authors assess for heterogeneity?
Discussion
The first paragraph repeats information from the introduction. First and second paragraph are more suitable for introduction.
Ln. 182: this a protocol. Only practical or operational or any issues not covered in other sections can be included in the discussion.
Author Response
We would like to thank the reviewer for such constructive comments of our work. We address each in the attached documnt.

Reviewer 2 Report
In the submitted manuscript, the authors present a protocol for a systematic review and meta-analyses, which aims to describe and estimate the relationship between participation in daily life activities and physical activity in stroke survivors. The review is registered in PROSPERO database. Comments for improving manuscript:
1) Please clearly identify one or more key questions which you aim to answer. Put these key question/s under a separate subheading.
2) Please add a PICOTS table with inclusion & exclusion criteria.
3) Please add evaluation of risk of bias (e.g. RoB and ROBINS-I tools) and strength of evidence (e.g. GRADE criteria using GRADEpro)
4) Please describe in greater detail how you will handle heterogeneity in stroke patients depending on area/region and extent of damage of parenchyma ? i.e. stroke patients with paralysis? patients who had post-stroke rehabilitation (management varies from country to country)?
5) Physical activity is measured in several different ways. Please enumerate the types and measurements included in your review: e.g. MVPA? LTPA? 24-hour physical activity levels? Will the unit be MET-minutes per week? or some other measure? In your search strategy, I see a wide variety of PA measures. It might be more practical to narrow your scope by selecting the most important/widely used measures.
6) Similar to PA, even activities of daily living might be recorded in different ways. Please enumerate example of what measures might be included: e.g. ADL functional scale? In your search strategy, I see a wide variety of participation measures. It might be more practical to narrow your scope by selecting the most important/widely used measures.
7) It is unclear to this reviewer, how results from this review might be useful in practice. Please describe the implications with more specificity/in more detail.
Points #4, #5, and #6 above can be incorporated into the PICOTS table as mentioned in point #2.
N/A
Author Response
We greatly appreciate your feedback and suggestions for improvement. We have addressed each comment in the attached document.

Reviewer 3 Report
The manuscript suggests a protocol for a systematic review and meta-analysis of observational studies regarding the relationship between participation in daily life activities and physical activity in stroke survivors. The importance of the intended study is explained well and the method or protocol is scientific and sound. But, the manuscript does not include any result of review or meta-analysis that the reviewer recommend to resubmit the paper after conducting the systematic review and meta-analysis with appropriate results.
Author Response
We are grateful for the feedback and time dedicated to review the manuscript. We have addressed your comments in the attached manuscript.

Round 2
Reviewer 1 Report
I congratulate and thank the authors for their attention to the suggestions and for carefully making the changes.
Reviewer 3 Report
All the comments of the reviewers were reflected well enough to be published in the journal.